# Federated epidemic surveillance

**Ruiqi Lyu** [iD] [1]*, **Roni Rosenfeld**[2], **Bryan Wilder**[2]

**1** Computational Biology Department, Carnegie Mellon University, Pittsburgh, Pennsylvania,
United States of America, **2** Machine Learning Department, Carnegie Mellon University, Pittsburgh,
Pennsylvania, United States of America

* ruiqil@cs.cmu.edu

Federated epidemic surveillance. PLoS Comput
Biol 21(4): e1012907. https://doi.org/10.1371/
journal.pcbi.1012907

Washington, UNITED STATES OF AMERICA

**Peer Review History:** PLOS recognizes the
benefits of transparency in the peer review
process; therefore, we enable the publication of
all of the content of peer review and author
responses alongside final, published articles.
The editorial history of this article is available
here: https://doi.org/10.1371/journal.pcbi.
1012907

**Data availability statement:** The data was
published at Delphi Epidata
https://cmu-delphi.github.io/delphi-epidata/.
The implementation of the experiments can be
found at https://github.com/Rachel-Lyu/
FederatedEpidemicSurveillance.

## Abstract

Epidemic surveillance is a challenging task, especially when crucial data is fragmented
across institutions and data custodians are unable or unwilling to share it. This study
aims to explore the feasibility of a simple *federated surveillance* approach. We con-
duct hypothesis tests on count data behind each custodian's firewall and then combine
*p*-values from these tests using techniques from meta-analysis. We propose a hypoth-
esis testing framework to identify surges in epidemic-related data streams and conduct
experiments on real and semi-synthetic data to assess the power of different *p*-value
combination methods to detect surges without needing to combine or share the underly-
ing counts. Our findings show that relatively simple combination methods achieve a high
degree of fidelity and suggest that infectious disease outbreaks can be detected without
needing to share or even aggregate data across institutions.

## Author summary

Timely and trustworthy epidemic surveillance requires data which is often fragmented
across institutions that cannot or will not share it for reasons of privacy, regulatory
restrictions, competition, etc. We show that infectious disease outbreaks can be detected
without sharing any raw data at all. To accomplish this, we introduce federated surveil-
lance: a method for pushing the computation behind these custodians' firewalls,
identifying sufficient statistics that can be shared to test for the presence of an out-
break without requiring access to any sensitive information. Across a variety of settings,
federated surveillance provides nearly the same ability to detect outbreaks as fully cen-
tralized data, and significantly more than could be achieved via any single data source
in isolation. Our results show this more readily implementable form of data sharing can
provide substantial value for future pandemic preparedness.

## Introduction

The prompt detection of outbreaks is critical for public health authorities to take timely and
effective measures. Providing early warning regarding either the emergence of a new pathogen
or a renewed wave of an existing epidemic allows for preparatory action to reduce transmis-
sion and prepare for increased load on the healthcare system. However, real-time surveillance

**Funding:** The author(s) received no specific funding for this work.

**Competing interests:** The authors have declared that no competing interests exist.

is challenging, particularly in countries such as the United States where relevant data is typically held by many separate entities such as hospitals, laboratories, insurers and local governments. These entities are often unable or unwilling to routinely share even aggregated time series such as the total number of patients with a specific diagnosis. Even when sharing aggregates is permitted from a privacy perspective (e.g., such disclosure is often allowable under U.S. HIPAA rules), a number of other barriers can arise due to competitiveness, commercial value, reputation, and other sources of institutional reluctance. For example, absolute numbers may be viewed as propriety if they are reflective of market share, or may be thought to reveal unwanted information about the relative performance of different institutions. Accordingly, public health authorities must mandate reporting for particular conditions of interest to create effective surveillance pipelines. This process is both cumbersome and reactive: a new reporting pipeline cannot be created until well into a public health emergency.

We propose and evaluate the feasibility of an alternative approach that we refer to as *federated epidemic surveillance*. The core concept is that health information, including even aggregate counts, never leaves the systems of individual data custodians. Rather, each custodian shares only specified *statistics* of their data, for example, the *p*-value from a specified hypothesis test. These statistics are then aggregated to detect trends that represent potential new outbreaks. Leveraging inputs from a variety of data custodians provides significantly improved statistical power: trends that are only weakly evident in any individual dataset may be much more apparent when the evidence is pooled together. To illustrate, consider COVID-19 hospitalizations in Seattle reported by four facilities to the US Department of Health & Human Services (HHS), as shown in Fig 1. As the patterns observed at different facilities vary substantially, it would be difficult to catch the overall trend by looking at any single facility. However, if the combined data from all facilities are available, a rapid increase in hospitalizations is clearly visible starting in March. Our goal is to detect outbreaks with comparable statistical power as if the data could be pooled together, but without individual data providers disclosing their time series of counts.

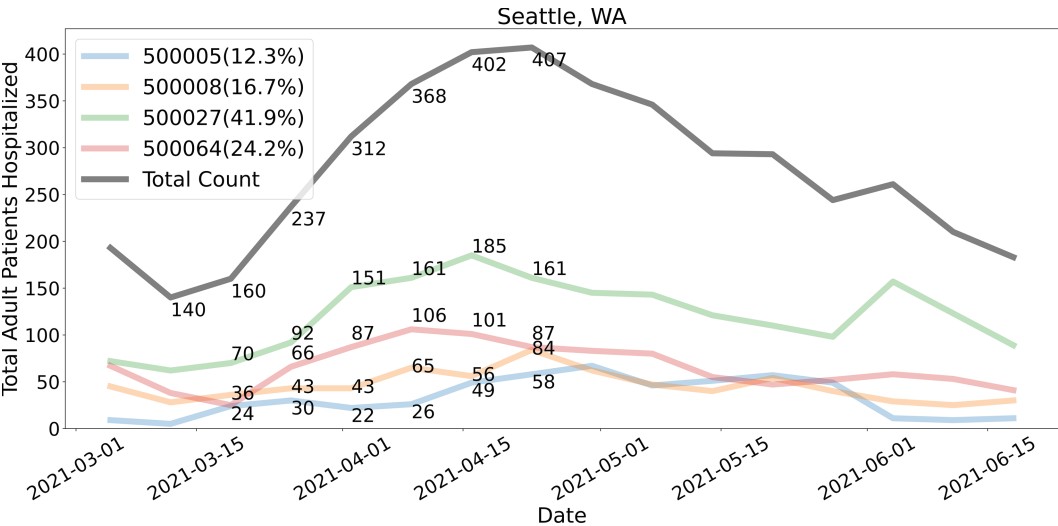

**Fig 1. Inpatient counts in Seattle, WA.** Adult patients hospitalized with confirmed COVID (7 day sum) of total and four largest facilities that together account for 95.12% of the market share in Seattle.

Our analysis shows that federated surveillance is indeed possible, often attaining performance similar to that with fully centralized data. We analyze a simple two-step approach: first, conduct individual hypothesis tests on the occurrence of a "surge" at different sites and subsequently use a meta-analysis framework to combine the resulting *p*-values into a single hypothesis test for an outbreak. More elaborate approaches (e.g. based on homomorphic computation or other cryptographic techniques) could allow more sophisticated computations under strong privacy guarantees. However, our goal is to demonstrate that high-performance federated surveillance is achievable using simple, easily explainable methods, since explainability to lay audiences is crucial for engendering trust and gaining acceptance. Our results indicate that effective epidemic surveillance is possible in environments with decentralized data, suggesting federated surveillance as a potential step towards modernizing surveillance systems in preparation for current and future public health threats. The implementation of the experiments can be found at https://github.com/Rachel-Lyu/FederatedEpidemicSurveillance.

## Materials and methods

### Datasets

We utilized two datasets for our analysis: (1) The "COVID-19 Reported Patient Impact and Hospital Capacity by Facility" dataset, provided by the U.S. Department of Health & Human Services, obtained through the Delphi Epidata API (https://cmu-delphi.github.io/delphi-epidata/)[1–3], covers the period from 2020-07-10 to 2023-03-03. This dataset provides facility-level inpatient counts on a weekly basis and primarily includes the "total adult patients hospitalized" metric. (2) The "Counts of claims with confirmed COVID-19" dataset is provided by Change Healthcare and covers the period from August 2, 2020 to July 30, 2022. This dataset provides county-level claim counts on a daily basis. We accumulate facility-level hospitalization counts to consider metro-area-level detection, and county-level insurance claim counts for state-level detection.

   We combine the summary statistics from 769 facilities from 286 metro-areas in the hospitalization dataset, and the summary statistics from 427 counties from 39 states in the insurance claim dataset. For decentralized analysis, we filter hospitalization summary statistics with more than 10% missing or 0 values, and insurance claim summary statistics with more than 50% missing or 0 values. We note these facilities still play a role in constructing the centralized ground truth. We also do not analyze single-site regions, as the decentralized and centralized statistics are the same.

   The hospitalization data tends to have larger counts distributed in fewer sites, while the claim data has smaller counts distributed in more sites. The level of imbalance is quantified using the normalized entropy metric

$$S = \frac{-\sum_{i=1}^{N} s_i \log s_i}{\log N},$$

where a value of 1 indicates perfectly equal shares. Fig 2 illustrates the differences between the two datasets in terms of the imbalance and the average reported counts magnitudes.

### Problem formulation

We explore the potential for simple federated surveillance methods to detect surges in a condition of interest using a variety of real and semi-synthetic data. We evaluate the success of our proposed framework by its ability to reconstruct the signal present in the fully centralized

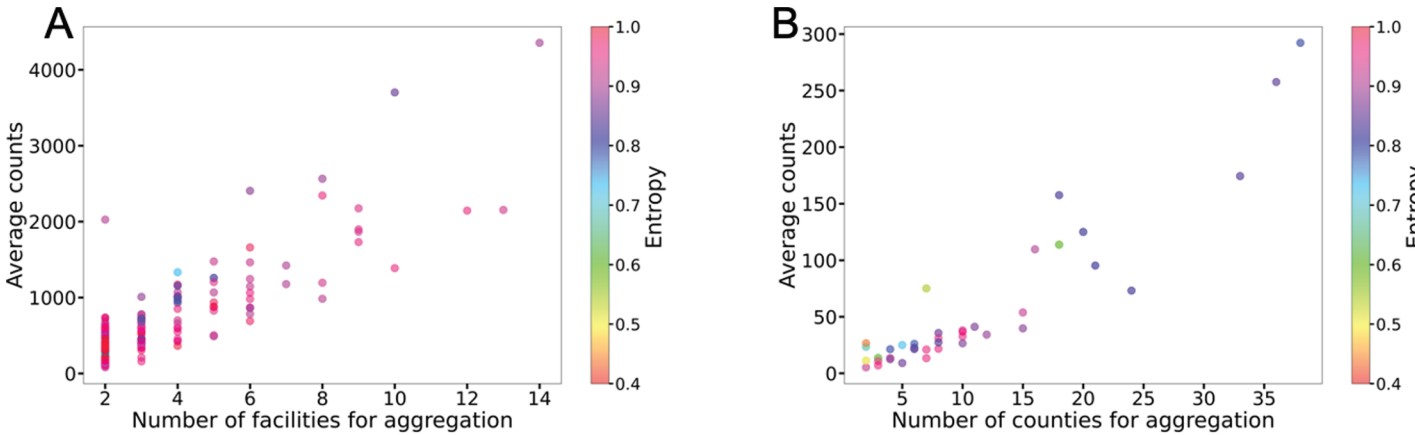

**Fig 2. The imbalance and the average reported counts magnitudes of the real data.** **(A)** Total adult patients hospitalized. **(B)** Counts of claims with confirmed COVID. Each dot is a region. The x-axes indicate how many facilities/counties are available for aggregation in that region. The y-axes indicate the average aggregated total count across time within a region.

setting, where all count information is available, or by recapitulating the signals indicated by semi-synthetic growth rates.

To start, we more formally introduce our objective. Precisely defining what constitutes a surge or outbreak is difficult. We operationalize a surge as a sufficiently large increase in the rate of new cases over a specified length of time. Formally, we model the time series $k_t$ of interest (e.g., cases or hospitalizations with a particular condition) as following a Poisson process $k_t \sim \mathrm{Poi}(\lambda_t)$ for some time-varying rate parameter $\lambda_t$. At a *testing* time $T$, we compare to a *baseline* period $B$, defined as $\{T - \ell, \ldots, T - 1\}$, and say that a surge occurs when the rate increases by at least a factor of $\theta$ (default 0.3) during the testing period compared to the baseline period. For simplicity, we model counts in the baseline period as following a Poisson distribution with a constant parameter $\lambda_B$: $k_{Bj} \sim \mathrm{Poi}(\lambda_B), j = T - \ell, \ldots, T - 1$. Similarly, during the testing period, we model $k_T \sim \mathrm{Poi}(\lambda_T)$ for a new parameter $\lambda_T$. We say that a surge occurs when $\lambda_T / \lambda_B > 1 + \theta$. We will analyze methods that test this hypothesis using the realized time series $k_t$, effectively asking whether a rise in counts must be attributed to a rise in the rate of new cases or whether it could be explained by Poisson-distributed noise in observations instead. A concise list of all the notations employed is provided in *"Parameters and notations"* section of S1 Text. Importantly, none of our results rely on the assumption that the data actually follows this generative process; indeed, we will evaluate using real epidemiological time series where such assumptions are not satisfied. Rather, our aim is to show that decentralized versions of this simplified hypothesis test can successfully detect surges. An overview of this framework is described in Fig 3.

Formally, we test the null hypothesis that the Poisson rate ratio $\lambda_T / \lambda_B$ is not larger than $1 + \theta$. We apply the uniformly most powerful (UMP) unbiased test for this hypothesis [4,5] with $p$-value $\Pr[r \geq k_T]$, where $r$ is a Binomial random variable $r \sim \mathrm{Bin}\left(\sum_{j=T-\ell}^{T-1} k_{Bj} + k_T, (1+\theta)/(1+\theta+\ell)\right)$. That is, to calculate the $p$-value of the Binomial test, we sum up the probabilities of observing more extreme values than $k_T$ if counts were uniformly split between the baseline and test periods. Of note, in this paper, we only discuss the unadjusted $p$-values. However, in practical applications, controlling the False Discovery Rate (FDR) of online multiple tests over time is crucial, while the consideration regarding $p$-value correcting and thresholding is a separate topic which we can refer to other articles like [6]. "Poisson rate ratio

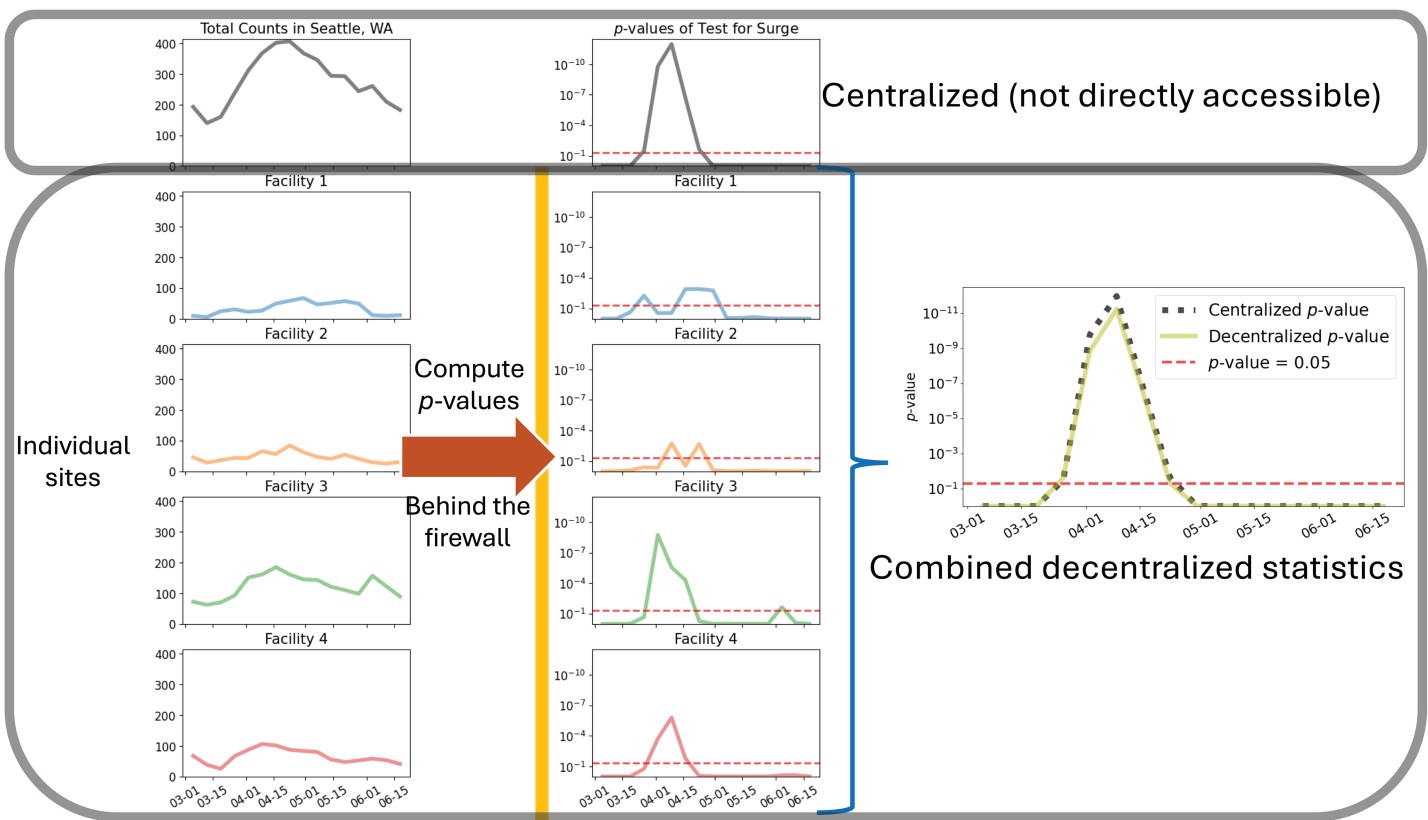

**Fig 3. Decentralized framework overview.** Hospitals share p-values, as opposed to raw count data, which are then combined with a unified hypothesis test.

**Table 1. Common meta-analysis methods.**

| Methods | Statistics | Distributions under the null |
|---|---|---|
| Stouffer's | $\sum_{i=1}^{N} \Phi^{-1}(p_i)$ | $N(0, N)$ |
| Fisher's | $-2\sum_{i=1}^{N} \log p_i$ | $\chi_{2N}^2$ |
| Pearson's | $-2\sum_{i=1}^{N} \log(1 - p_i)$ | $\chi_{2N}^2$ |
| Tippett's | $min\{p_1, \ldots, p_N\}$ | $Beta(1, N)$ |

test for detecting a surge" in the Materials and methods section includes more details about the hypothesis test.

In the federated setting, each data custodian computes $p$-values for this hypothesis test using only their own time series counts. The $p$-values are then combined using methods from meta-analysis (see "Overview of meta-analysis methods" in Materials and methods section for more discussions). Considering $p_1, \ldots, p_N$ are $p$-values obtained from $N$ independent hypothesis tests and the joint null hypothesis for the $p$-values is $H_0 : p_i \sim U[0, 1], i = 1, \ldots, N$ [7], several commonly used statistics and their corresponding distributions can be computed accordingly. We listed some popular ones in Table 1.

## Poisson rate ratio test for detecting a surge

Our model assumes that in a short time period such as a week or a month, if there is no surge, the counts follow a Poisson distribution with a rate parameter $\lambda$, which can be estimated

based on observations from the previous period. However, in the presence of a sudden surge, the distribution changes, and the rate parameter $\lambda$ increases.

To identify surges, we propose conducting a hypothesis test to determine whether the increase in Poisson rates exceeds a user-defined threshold, denoted as $\theta$. This threshold can be tailored to the inherent characteristics of different indicators, allowing for an adjustable tradeoff between sensitivity and specificity. Specifically, a surge is defined as a Poisson rate that increases by a factor of at least $\theta$ during the testing period with Poisson rate parameter $\lambda_T$, compared to the baseline period with parameter $\lambda_B$. Formally, we test the null hypothesis

$$H_0 : \frac{\lambda_T}{\lambda_B} \leq 1 + \theta. \tag{1}$$

We propose utilizing the UMP unbiased test, which is a method of testing two Poisson rates ratio first proposed by Przyborowski and Wilenski [8]. This test is based on conditioning on the summation of the counts of the whole period including the baseline period and testing period. We consider $k_{Bj}, j = T-\ell, \ldots, T-1$ and $k_T$ as independently distributed according to $\text{Poi}(\lambda_B)$ and $\text{Poi}(\lambda_T)$, so that their joint distribution can be written as

$$P(k_B, k_T) = \frac{e^{\ell\lambda_B + \lambda_T}}{k_{B(T-\ell)}!\ldots k_{B(T-1)}!k_T} \exp\left[ k_T \log \frac{\lambda_T}{\lambda_B} + \left( \sum_{j=T-\ell}^{T-1} k_B + k_T \right) \log \lambda_B \right] \tag{2}$$

By Theorem 4.4.1 in [4], there exist UMP unbiased tests concerning the ratio $\lambda_T/\lambda_B$. By the Theorem, the tests are performed conditionally on the integer points of the hyperplane segment, which is the total counts over both periods equals to $\sum_{j=T-\ell}^{T-1} k_B + k_T$, in the positive quadrant of the $(k_{B(T-\ell)}, \ldots, k_{B(T-1)}, k_T)$ space. The conditional distribution of $k_T$ given total counts is the binomial distribution corresponding to $\sum_{j=T-\ell}^{T-1} k_B + k_T$ trials and probability $(1 + \theta)/(1 + \theta + \ell)$ of success. Then the UMP unbiased test can be written as Equation 3.

$$H_0' : \frac{\lambda_T}{\lambda_T + \ell\lambda_B} \leq \frac{1 + \theta}{1 + \theta + \ell} \tag{3}$$

Essentially, the UMP test corresponds to a Binomial test that examines the indicator during the testing period conditioning on the total counts of both the baseline and testing periods. Using this test, we can easily determine the $p$-value through a one-tailed exact test. The formulation of the $p$-value is shown as Equation 4.

$$p = \Pr\left( r \geq k_T | \sum_{j=T-\ell}^{T-1} k_{Bj} + k_T, \frac{1 + \theta}{1 + \theta + \ell} \right) = \sum_{r=0}^{c} \binom{n}{r}(1 - \rho)^{n-r}\rho^r \tag{4}$$

where $p$-value is the probability of count $r$ being greater than or equal to $k_T$ given the total counts over the baseline period and testing period, and the summation in the second line for count $r$ from 0 to $c := \sum_{j=T-\ell}^{T-1} k_{Bj}, n := \sum_{j=T-\ell}^{T-1} k_{Bj} + k_T, \rho := \ell/(1 + \theta + \ell)$. This test is known for being exact while conservative, as the actual significance level is always below the nominal level [9].

The power of a hypothesis test is defined as the probability of rejecting the null hypothesis when the alternative hypothesis is in fact true. In the case of the Binomial test under the null hypothesis, we need to determine the critical value $k_{cr}$ for $k_T$ at a given type I error rate $\alpha$. This critical value represents the minimum number of successes in the

sample required to reject the null hypothesis in favor of the alternative hypothesis. Mathematically, the critical value is determined by finding $k_{cr}$ that satisfies the condition $\Pr(r \geq k_{cr} | n = \sum_{j=T-\ell}^{T-1} k_{Bj} + k_T, (1+\theta)/(1+\theta + \ell)) \leq \alpha$. Under the alternative hypothesis, characterized by a higher growth rate of the Poisson rate $\theta' > \theta$, the power is computed as $\Pr\left(r \geq k_{cr} | n = \sum_{j=T-\ell}^{T-1} k_{Bj} + k_T, (1+\theta')/(1+\theta' + \ell)\right)$. The power quantifies the test's ability to detect a surge when it truly exists.

The analytical formula for calculating power in the discrete distribution makes it difficult to see the contribution of different parameters from its form directly, as the critical value and the counts should be integers. However, in the figures of the power analysis, we used the exact power rather than the approximation. As a practical alternative, a Gaussian approximation of Binomial distribution with continuity correction can be employed. This approximation will ignore the rounding errors on the power calculation by allowing for the decimals in the values while maintaining an acceptable level of precision. With continuity correction, we can compute the power as Equation 5. The details of the proof are in *"Power computation"* section of S1 Text.

$$\text{power} = \Phi\left( \frac{\sqrt{n\ell}(\theta' - \theta)}{(1+\theta+\ell)\sqrt{(1+\theta')}} - \frac{Z_\alpha(1+\theta'+\ell)\sqrt{1+\theta}}{(1+\theta+\ell)\sqrt{(1+\theta')}} - \frac{1+\theta'+\ell}{2\sqrt{n\ell(1+\theta')}} \right) \tag{5}$$

The expression inside the $\Phi(\cdot)$ function comprises three terms, each capturing a specific aspect of the analysis. The first term quantifies the impact of the total counts' magnitude, while the second term relates to the type I error rate. The third term corresponds to the continuity correction term, which can be ignored when the sample size $n$ is sufficiently large. This formula provides an approximation of the power and allows for a more intuitive understanding of the influence of different parameters.

## Overview of meta-analysis methods

Meta-analysis is known for its ability to enhance statistical power by combining signals of moderate significance, effectively controlling false positives, and enabling comparisons and contrasts across tests and time [10]. The properties of various $p$-value combination methods have been extensively studied. For instance, Heard and Rubin-Delanchy [7] observed that Tippett's and Fisher's methods are more sensitive to smaller $p$-values, while Pearson's methods are more sensitive to larger $p$-values. They also suggested that Fisher's and Pearson's methods are more suitable for testing positive-valued data under the alternative hypothesis, with Fisher's method performing better for larger values and Pearson's method for smaller values. Additionally, Stouffer's method is often preferred for testing real-valued data that approximates a Gaussian distribution.

Among the various combination methods, Stouffer's and Fisher's methods have gained significant attention in the literature due to their popularity in meta-analysis. Elston [11] noted that when the number of sites is very large, Fisher's method will give a combined $p$-value that is close to 0 when the actual $p$-value is below $1/e$; and will give a combined $p$-value that is close to 1 when the actual $p$-value is above $1/e$. Similarly, Stouffer's method has the threshold point as $1/2$, while Pearson's method has $1 - 1/e$. Rice [12] suggested that methods like Stouffer's are more appropriate when all tests are homogeneous and the combined $p$-value can be interpreted as a "consensus $p$-value". On the other hand, Fisher's method is particularly useful when testing against broad alternatives which specifically tests whether at least one component test is significant. It has shown superiority in certain scenarios, such as in GWAS where there may be significant differences in effect sizes between different populations [10].

In such cases, Stouffer's and Lancaster's [13] methods tend to lose power where there are only a few studies showing strong evidence of rejecting the hypothesis. Fisher's method's has an advantage in handling potential negative correlations between entities, which can arise due to factors like competition.

The meta-analysis methods outlined in Table 1 are based on the assumption that test statistics adhere to continuous distributions, with the joint null hypothesis for the $p$-values being $H_0 : p_i \sim U[0, 1], i = 1, \ldots, N$ [7]. However, this conventional understanding of $p$-value distributions doesn't align with the hypothesis test we introduce in our study. There are several reasons contributing to this discrepancy. First, for discrete distributions, such as the Binomial, $p$-values aren't uniformly distributed under the null hypothesis due to truncation. Second, given the target as the centralized counts, the decentralized counts exhibit a negative correlation, making the dependence structure for decentralized $p$-values more complicated. Moreover, when the null hypothesis is rejected, the distribution of $p$-values undergoes a shift, necessitating strategies to mitigate the resulting decrease in power during meta-analysis. To address these issues, we present a perspective from the explicit mathematical format of the $p$-values in both centralized and distributed settings, followed by a discussion on how to combine them with minimal loss.

In the Binomial test of the distributed setting, the $p$-value for site $i$ can be expressed as Equation 6:

$$p_i = \Pr(r \geq k_{Ti} | \sum_{j=T-\ell}^{T-1} k_{Bij} + k_{Ti}, \frac{1+\theta}{1+\theta+\ell}) = \sum_{r=0}^{c_i} \binom{n_i}{r}(1-\rho)^{n_i-r}\rho^r \qquad (6)$$

where $p$-value for site $i$ is the probability of count $r$ being greater than or equal to $k_{Ti}$ given the total counts of that site over the baseline period and testing period, and the summation in the second line is calculated for count $r$ from 0 to where $c_i := \sum_{j=T-\ell}^{T-1} k_{Bij}, n_i := \sum_{j=T-\ell}^{T-1} k_{Bij} + k_{Ti}$.

It is evident that directly combining the $p$-values of different sites from Equation 6 to Equation 5 without any loss is not feasible, as we need to deal with n choose r, but we have no access to the counts themselves. Therefore, our goal lies in finding better approximations and developing improved methods for combining the approximations, trying our best to use less count information. The selection among various meta-analysis methodologies mirrors the quest for a more precise approximation. We will further explore combination techniques, emphasizing Stouffer's and Fisher's methods, and elucidate how mathematical formulations assist in prudently integrating auxiliary information. For instance, we might consider the estimated contributions from different data providers or attribute weights to studies. Weighting stands as a prevalent strategy for assimilating evidence, while discerning the optimal weighting scheme also demands consideration. Beyond introducing weights, Stouffer's method can be further refined given estimates of the aggregate reports from all sites $n = \sum_{i=1}^{N} n_i$. This refinement introduces a continuity correction term, tempering over-conservative results when the overall counts are small.

## Stouffer's method

Stouffer's method is employed by utilizing the Gaussian approximation of the Binomial parameter, which is based on the central limit theorem. This approach is commonly used when analyzing the Binomial and Poisson distributions, especially when the counts are sufficiently large. To test the success probability $\rho$ using Stouffer's method, the distribution of $c/n - \rho$ is approximated by $N(0, \rho(1-\rho)/n)$. The z-score can then be calculated as $z = (c - n\rho)/\sqrt{n\rho(1-\rho)}$ [14]. The $p$-value of the Binomial exact test is determined by the

cumulative distribution function $F_{\text{Bin}}(c; n, \rho)$. Define rounding error or fluctuation term $\epsilon_r = 1/2 - \{(n\rho + z\sqrt{n\rho(1-\rho)}) - \lfloor(n\rho + z\sqrt{n\rho(1-\rho)})\rfloor\}$, which takes values in the interval $[-1/2, 1/2]$. Thus, the true $p$-value in terms of the z-score with error terms can be expressed as Equation 7 [14–16].

$$p = \Phi(z) + \left( \frac{(1-2\rho)(1-z^2)}{6} + \epsilon_r \right) \frac{\Phi(z)}{\sqrt{n\rho(1-\rho)}} + \mathbf{O}(n^{-1}) \tag{7}$$

It should be noted that the denominator $\sqrt{n\rho(1-\rho)}$ in the first-order error term indicates that Stouffer's method may be unreliable for small sample sizes or when the probability is close to 0 or 1 [17].

After applying the Gaussian approximations, the combination of $p$-values becomes the next focus. One limitation of the naive meta-analysis methods is the assumption of equal contributions across studies, which may not hold true, especially when the studies have significantly different sizes. Determining appropriate weights for different studies poses a challenging task. In the case of Stouffer's method, some studies in GWAS suggest using the inverse of the standard error or the square root of the sample size as weights [18]. Our proposed test is a special case of meta-analysis, as the centralized counts are the summation of decentralized counts. In this case, we can combine the approximations without any loss by introducing weights. By obtaining a centralized $p$-value $p$ approximates $\Phi\left( (\sum_{i=1}^{N} c_i - \rho \sum_{i=1}^{N} n_i)/\sqrt{\rho(1-\rho) \sum_{i=1}^{N} n_i} \right)$ using distributed $p$-value $p_i$ approximates $\Phi\left( (c_i - \rho n_i)/\sqrt{\rho(1-\rho)n_i} \right)$, it turns out that the weights for aggregating the $p$-values is the square root of the shares of each entity (estimated using auxiliary information or delayed counts report, see *"Enhancing federated surveillance with auxiliary information"* in *Results* section), which is formulated as Equation 8.

$$p = \Phi\left( \sum_{i=1}^{N} \sqrt{s_i} \Phi^{-1}(p_i) \right) \tag{8}$$

Furthermore, improvements can be made by incorporating a continuity correction when an estimated value for $n = \sum_{i=1}^{N} n_i$, representing the total counts of all entities, is available. Due to the discreteness of the Binomial distribution and the continuity of the normal distribution, the correction is helpful when $n$ is not sufficiently large. One commonly used correction is Yates' correction in the Binomial test, which involves subtracting $1/2$ from the absolute difference between the observed count $c$ and the expected count $n\rho$. Considering our case where $c < n\rho$, the approximation of the $p$-value can be rewritten as $\Phi\left( (c + 1/2 - \rho n)/\sqrt{\rho(1-\rho)n} \right)$.

Similarly, we can derive the combination formula as Equation 9.

$$p = \Phi\left( \sum_{i=1}^{N} \sqrt{s_i} \Phi^{-1}(p_i) + \frac{1-N}{2\sqrt{\rho(1-\rho)n}} \right) \tag{9}$$

The additional term $(1-N)/(2\sqrt{\rho(1-\rho)n})$ accounts for the effect of the continuity correction on the combined $p$-value. This correction becomes more necessary when the number of entities $N$ is large, but the total counts during the baseline and testing procedures are relatively small, indicating more dispersed data. In such cases, the correction term becomes more significant. If other types of coarse-grained evidence are available to estimate the counts' magnitudes, a correction term can be added to make the combined $p$-value less conservative.

## Fisher's method

The statistical tests based on Fisher's method and Pearson's method involve taking the logarithm of the $p$-values and summing them. The rationale behind the log sum approaches is that the $p$-value $F_{\text{Bin}}(c; n, \rho)$ is upper and lower bounded by exponential functions such as Equation 10. See *"Proof for Equation 10"* section of S1 Text for the details of the proof.

$$\frac{1}{\sqrt{2n}} \exp\left(-nD(\frac{c}{n}\|\rho)\right) \le p \le \exp\left(-nD(\frac{c}{n}\|\rho)\right) \tag{10}$$

where $D((c/n)\|\rho)$ represents the relative entropy (Kullback-Leibler divergence) between $(c/n, (n-c)/n)$ and $(\rho, 1-\rho)$, which is $(c/n)\log(c/(n\rho)) + ((n-c)/n)\log((n-c)/(n(1-\rho)))$.

The logarithm of the $p$-value are upper bounded by $-nD((c/n)\|\rho)$, allowing the summation of logarithmic $p$-values from different sources to be meaningful. The formula indicates that the choice of $\rho$ and $n$ is relatively flexible for Fisher's method, compared with the error term of Stouffer's method which is in proportion to $(n\rho(1-\rho))^{-1/2}$. Our experiments also support the idea that when the reported magnitude is small and the testing period is short, like the the "Counts of claims with confirmed COVID-19", Fisher's method is more reliable than Stouffer's.

Different weighting strategies for Fisher's method have been investigated, and various modifications have been proposed [10,13,19,20]. However, the optimal weighting scheme remains uncertain. Some studies suggest employing adaptively weighted statistics combined with permutation tests [21] or using Monte Carlo algorithms to approximate the rejection region and determine optimal weights. Another approach involves constructing Good's statistic [20], which is a weighted statistic defined as $-2\sum_{i=1}^{N} w_i \log p_i$ with weight $w_i$ for site $i$. Under the null hypothesis, it follows a chi-squared distribution with $2\sum_{i=1}^{N} w_i$ degrees of freedom (DF). Here, we use the weighting scheme $w_i = s_i N$, where $s_i$ represents the share of each site. This weighting ensures that the resulting chi-squared statistic has a total DF equal to $2N$, i.e., $-2\sum_{i=1}^{N} s_i N \log p_i \sim \chi^2_{2N}$.

Additionally, some methods leverage the fact that the $1-p$ quantile of the Gamma distribution $\text{Gam}(\alpha=1, \beta)$ is $-\log p/\beta$, i.e., $F_{\text{Gam}(\alpha=1,\beta)}(-\log p/\beta) = 1-p$, where $\beta = 1/2$ represents Fisher's methods. For example, Lancaster's method [13] sets $\beta = 1/2$ and transforms each $p_i$ to the $1-p_i$th quantile of the Gamma distribution with $\alpha = s_i/2$. This transformation yields $X_i = F^{-1}_{\text{Gam}(s_i/2,1/2)}(1-p_i) \sim \chi^2_{s_i}$. By additivity, we have $\sum_{i=1}^{N} X_i \sim \chi^2_{\sum_{i=1}^{N} s_i}$. In summary, Lancaster's method generalizes Fisher's method by assigning different weights to the DF of each source, resulting in a larger total DF compared to Fisher's method. However, Yoon et al. [10] demonstrated that the large DF cause the individual distributions to approach the normal distribution, leading to a significant decrease in power. Yoon et al. consequently proposed the *wFisher* method, which employs a similar weighting scheme but shrinks the total DFs to match those of the original Fisher's method. Specifically, they constructed the statistics $\sum_{i=1}^{N} F^{-1}_{\text{Gam}(w_i N/2,1/2)}(1-p_i) \sim \chi^2_{2N}$. We observe that the *wFisher* method exhibits greater stability compared to other weighting methods. The *wFisher* framework can be written as Equation 11.

$$p = 1 - F_{\chi^2_{2N}}\left(\sum_{i=1}^{N} F^{-1}_{\text{Gam}(\frac{s_i N}{2}, \frac{1}{2})}(1-p_i)\right) \tag{11}$$

### Evaluation of the surge detection task

In the surge detection task, our evaluation centers on the timeliness of detecting surges, which is represented by binary sequences denoting the presence or absence of a surge. Alerts based on $p$-values are triggered when these values dip below a predetermined threshold, while growth rate alerts are activated when the growth rate of the Poisson rate parameter exceeds a set threshold. For real data analysis, the $p$-value alerts from the centralized setting serve as the ground truth. In contrast, the semi-synthetic data analysis uses the growth rate alerts as the ground truth. Once the ground truth is established, both centralized and decentralized $p$-value alerts are assessed by comparing them to this benchmark.

For each ground truth alert, the reconstructed alerts are considered true positives if they fall within a specified time window, e.g., no earlier than one week before and no later than two weeks after. Otherwise, the reconstructed alerts are classified as false positives. Moreover, any true alerts not matched by the constructed alerts are deemed false negatives. Following this rule, Precision and Recall metrics can be calculated. Precision represents the ratio of true positives (TP) to the summation of true positives and false positives (FP) (TP/(TP + FP)). Recall, on the other hand, denotes the ratio of true positives to the summation of true positives and false negatives (FN) (TP/(TP + FN)). These metrics are computed for different confidence level thresholds. Finally, the Precision-Recall metric is obtained, and the power (equal to Recall) is evaluated while controlling the FDR, which equals 1 - Precision, at 0.10, allowing for an assessment of method performance. It should be noted that the term "power" in this context refers to the power of the classification task, as opposed to the power associated with a Binomial test that was mentioned earlier.

The aggregate level "true" surges vary a lot in the datasets of our experiments, and we filtered those series with no surges out of our analysis. For the hospitalization dataset, the mean and the standard deviation of the number of surges in each metro-area-level series are 15.82 and 3.87; for insurance claim dataset, the mean and the standard deviation in each state-level series are 19.51 and 16.20. More detailed descriptions of ground truth windows are described in *"Summary of ground truth windows"* section of S1 Text.

### Semi-synthetic data analysis

The semi-synthetic analysis is conducted under the assumption of noisy data, where the observed signal deviates from the true underlying prevalence. There are several objectives of this analysis. Firstly, it aims to investigate the effects of various factors, such as the number of sites, magnitudes of reports, and the imbalance of shares, while controlling for other dimensions. Secondly, the analysis facilitates the comparison of systematic errors arising from noise in the centralized and federated settings, as well as the assessment of the combination loss during the meta-analysis process. By utilizing real data as a starting point and employing semi-synthetic data analysis, we are able to control and examine the impact of different data features including the total count magnitudes and shares of entities.

The generation of the semi-synthetic data starts with the counts of outpatient insurance claims with a primary diagnosis of COVID-19 in each county provided by Change Healthcare, covering the period from 2020-08-02 to 2022-07-30 on a daily basis. Initially, a 7-day moving average smoother is applied to the total counts of all counties in a state. We then treat the resulting smoothed values as the underlying occurrence rate confirmed COVID-19 cases each day, from which Poisson-distributed observed counts are simulated. That is, Poisson sampling is performed to generate simulated observations, assuming that the observed data is drawn from a Poisson distribution with the smoothed data serving as the rate parameter. After getting the state-level counts, we multinomially allocate them into different sites

with the designed parameter, regarding them as count-level counts. Once the simulated counts are obtained, the next step involves computing the growth rate of the ground truth prevalence and the *p*-values for the hypothesis test at each time point. Subsequently, alerts are determined based on these computed values and the predetermined thresholds.

In contrast to the previous setting, where we established the ground truth through hypothesis testing in the centralized data, we now define it based on the growth of the Poisson rate surpassing a predetermined threshold. This shift is due to the gap between the Poisson rate used for simulation and the Poisson counts in the centralized setting. The discrepancy between the growth rate alert of the Poisson rate and the centralized *p*-value alert can be ascribed to the inherent properties of the Poisson assumption and the introduced Poisson rate ratio test. Additionally, the distinction between centralized and decentralized *p*-value alerts directly results from the meta-analysis procedure. By examining and contrasting the errors arising from this bifurcated process, we demonstrate that the recombination cost exerts a comparable influence on the performance as does systematic noise. Moreover, the weighted and corrected Stouffer's methods and the weighted Fisher's method exhibit stability across various settings using auxiliary information, illustrating their robustness in practical scenarios.

## Results

### Efficacy of federated surveillance

We start by studying the statistical power and sensitivity of federated surveillance methods compared to centralized data, i.e., whether decentralized hypothesis tests allow comparable accuracy in detecting surges compared to the (unattainable) ideal setting where all data could be pooled for a single test. We also compare performance to performing the test using only the largest site within a region. We assess decentralized methods using both their theoretical expected accuracy on data drawn from our simplified generative model and on two real COVID-19 datasets.

Fig 4 shows the expected statistical power of each meta-analysis method for combining *p*-values on data drawn from our generative model, compared to the statistical power of a centralized version of the same hypothesis test and to a version that uses only the counts from a single data provider. We fix a threshold of $\theta = 0.3$ for a surge. The *x* axis varies the true rate of growth, with a higher power to detect surges when they deviate more significantly from the null. To ensure a fair comparison, we calibrate the rejection threshold for each method to match the nominal $\alpha = 0.05$ rejection rate when the true growth rate is exactly 30% (i.e., precisely satisfying the null). To perform simulations, we set a total count over both training and testing periods to be 200. The counts are binomially simulated in the testing period with probability parameter $(1 + \theta')/(1 + \theta' + \ell)$, where $\theta'$ is the growth rate to be tested. The total and testing period counts are multinomially allocated between 2 sites (Fig 4A) and 8 sites (Fig 4B) with a uniform distribution across the sites (an assumption we will revisit later).

We find that, in this idealized setting, the top-performing federated method (Stouffer's method) almost exactly matches the power of the centralized data test. Conversely, significant power is lost by using only the *p*-value from a single site, indicating that sharing information across sites is necessary for good performance. The other meta-analysis methods exhibit lower power than Stouffer's; in later sections, we will examine the settings in which different meta-analysis methods for combining the *p*-values lead to better or worse performance. More experiments are shown in *"Power curves and calibrations"* section of S1 Text.

However, this idealized setting is highly simplified based on our assumptions. Real-world epidemic data deviates from such assumptions in multiple ways, including non-stationary

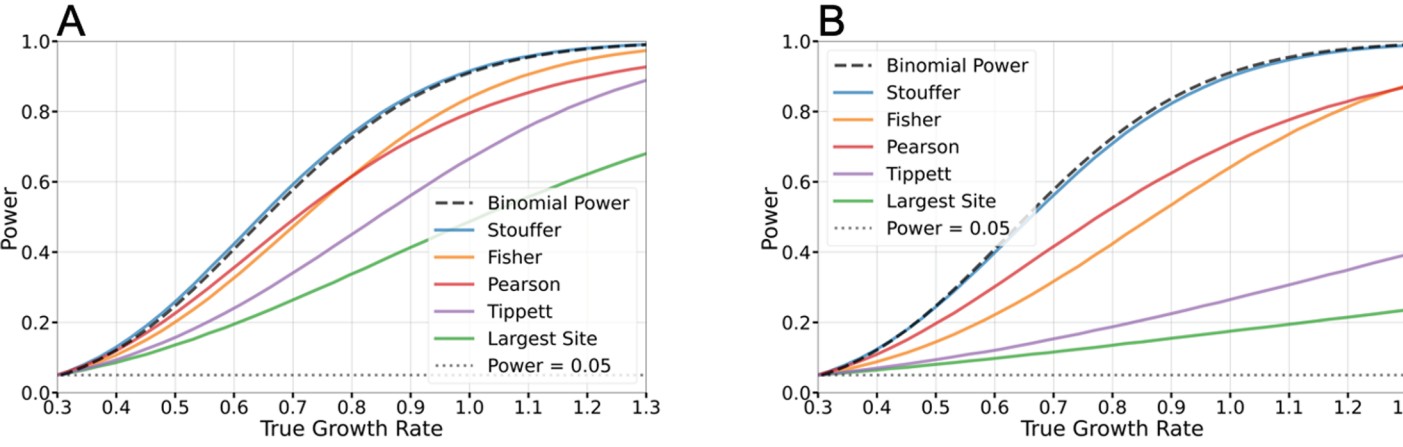

**Fig 4. Power analysis of federated surveillance methods. (A)** Two sites provide the *p*-values. **(B)** Eight sites provide the *p*-values.

time series and non-uniform distribution of patients among facilities. To validate the robustness of our federated surveillance framework in the real world, we use two datasets providing a more realistic representation of the complexities and challenges, allowing us to assess the performance of the methods under more diverse conditions. The first dataset is COVID-19 hospitalization reported in the "COVID-19 Reported Patient Impact and Hospital Capacity by Facility" covering the period from July 10, 2020 to March 3, 2023. The second dataset is a daily time series of the total number of outpatient insurance claims with a primary diagnosis of COVID-19 in each county covering the period from August 2, 2020 to July 30, 2022.

In our analysis of these datasets, we add up the counts of facility-level hospitalization to generate metro-area-level *p*-value alerts, and county-level insurance claim counts to create state-level *p*-value alerts. The term "alert" in this context indicates the occurrence of a significant increase. Formally, a *p*-value alert indicates that the confidence level of rejecting the null hypothesis is below a predetermined threshold $\alpha = 0.05$. For more details of evaluation, see *"Evaluation of the surge detection task"* in *Materials and methods* section. Due to variations in reporting frequency and the number of sites, the hospitalization data tends to have larger counts distributed in fewer sites, while the claim data has smaller counts distributed in more sites. By applying our methods to these datasets, we obtain the recall-precision curves as Fig 5. The recall and precision of the combined *p*-values are evaluated against the ground truth as the *p*-value alerts on the centralized data, i.e., the total counts in that geographic region. The results demonstrate that the federated test, with the appropriate combination method, can effectively reconstruct centralized information. In our analysis of facility-level hospitalization data, Stouffer's method achieved a recall of 0.95 when we set precision to 0.90, corresponding to a false discovery rate (FDR) of 0.10, and achieving an area under the curve (AUC) of 0.98. When examining county-level claim data, Fisher's method attained a recall of 0.76 at a precision of 0.90, and an AUC of 0.95. Using data only from the largest single facility produced lower accuracy. Specifically, for hospitalization and claim data, when fixing the precision to 0.90, their recall is both 0.62. Their AUCs are 0.88 and 0.90. In insurance claim data, we observe 71.1% and 88.9% of true positives detected within 3 days of the true surge for Stouffer's and Fisher's methods, respectively. As hospitalizations are reported weekly, we observe 98.4% and 99.2% of true positives detected at the week of the true surge for Stouffer's and Fisher's methods, respectively. Figures of the distribution of detection days or weeks are shown in *"Detection Delay"* section of S1 Text. Importantly, this data is directly drawn from

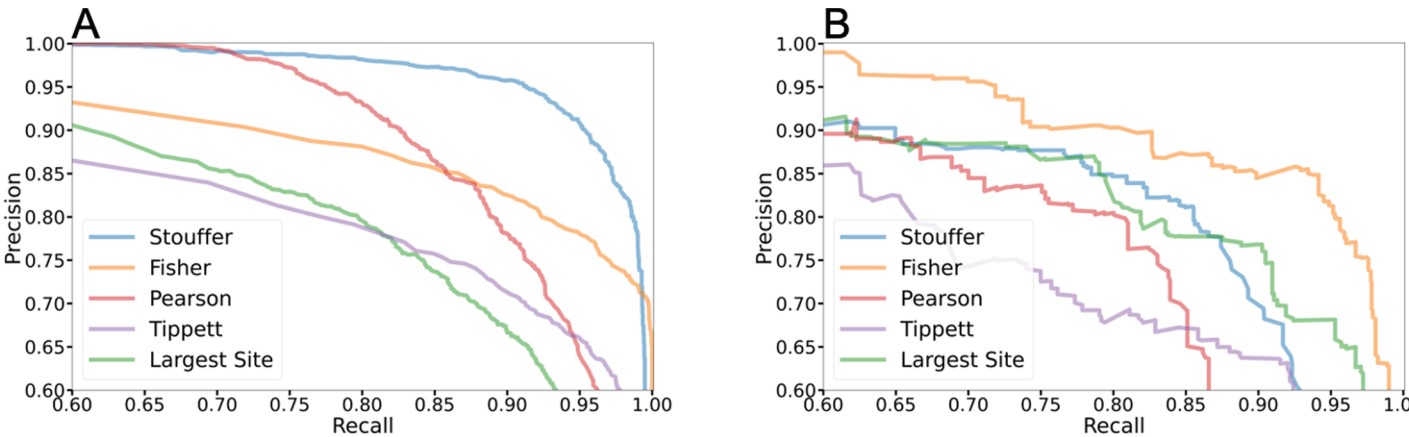

**Fig 5. Real data analysis of federated surveillance methods. (A)** Facility-level hospitalizations: *p*-values from weekly counts combined from multiple separate facilities to generate metro-area-level *p*-value alerts. **(B)** County-level insurance claims: *p*-values from daily counts combined from multiple separate counties to generate state-level *p*-value alerts.

the real world, and need not satisfy the assumptions of our generative process. The results demonstrate the potential of federated methods for early detection of outbreaks, even for *p*-values combined with only the simplest meta-analysis framework.

The selection of the highest-performing meta-analysis method should depend on the characteristics of the underlying time series. In the next subsection, we will use semi-synthetic data to explore how such characteristics impact the performance of the different meta-analysis methods.

## Comparative performance of the different combination methods

Based on our discussion on theoretical analysis and the experiments on the real-world data above, we have drawn preliminary conclusions regarding our proposed test. Firstly, employing a combined test using a meta-analysis framework generally yields superior performance compared to relying solely on a single entity, even if the entity contributes a significant portion of the counts. Secondly, the optimal choice of method depends on specific features of the data. If the reporting sites are comparable in size and the counts have relatively large magnitudes, Stouffer's method is preferable. However, if the sites' sizes are uneven, $\log p$-based Fisher's method provides superior performance [11] (Fig 6).

To better understand the impact of various aspects of the data on the combination process, we can selectively modify one factor while keeping others constant. Several factors might influence the success of federated surveillance methods. Firstly, as the number of reporting entities increases, combining *p*-values becomes increasingly complex. This arises because each entity introduces a blend of variabilities—changes in population size, prevalence, observation noise, and potential negative correlations between sites—that complicates the separation of distinct effects in a single *p*-value. In that case, both multiplicative and additive effects on the *p*-values and their approximations are amplified with more facilities, which leads to more biased results. Secondly, the magnitude of the counts affects which *p*-value combination method performs best, as evident from the power formula in Equation 5 in the Materials and methods section. Thirdly, the imbalance in the proportions of the data providers in terms of the cases in a region challenges the robustness of the combination methods.

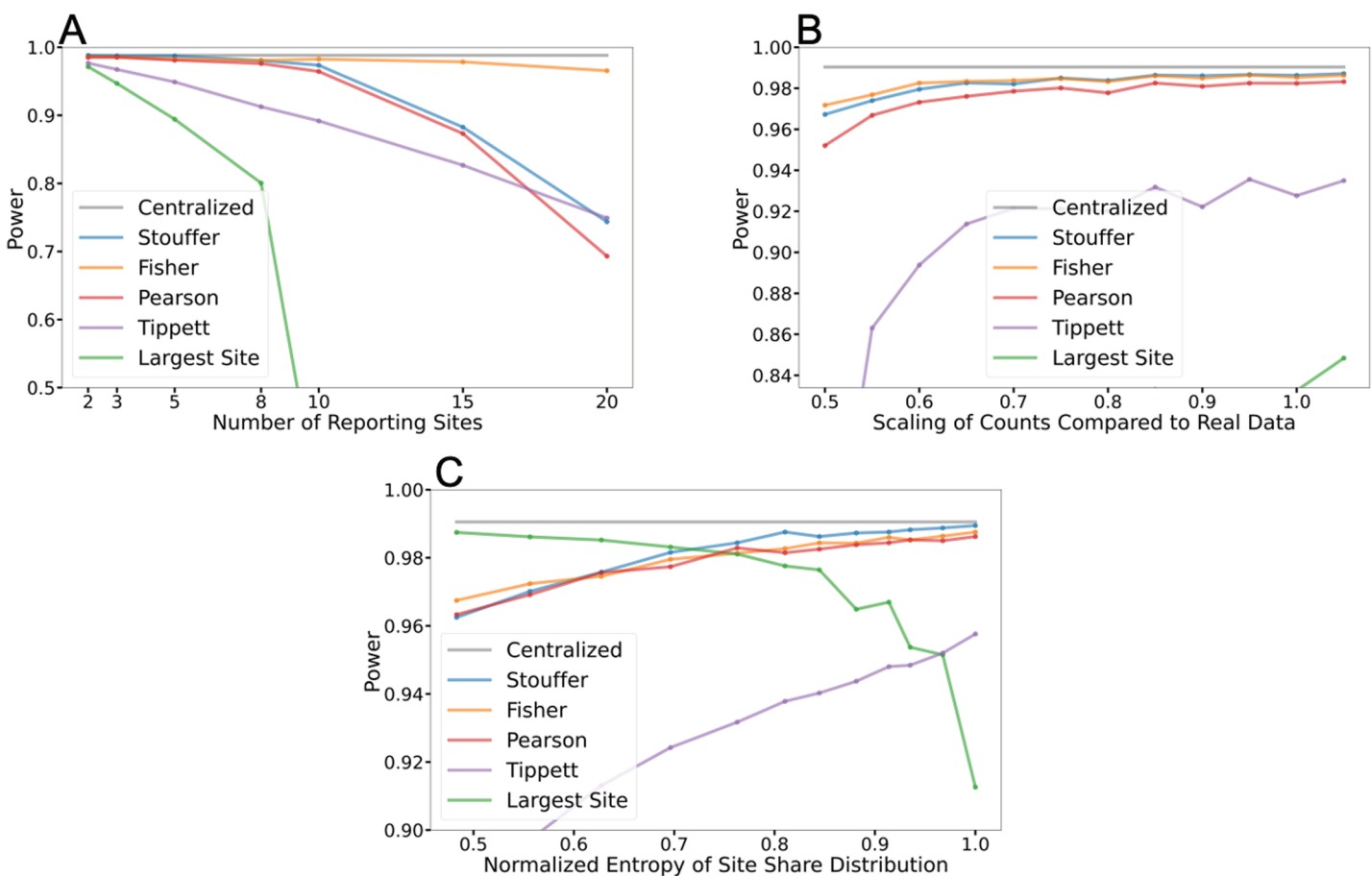

**Fig 6. Federated surveillance methods on semi-synthetic data, varying site-level data generating process. (A)** Numbers of reporting sites (of equal share). **(B)** Effect of varying the total magnitude of counts, equally split between eight sites. The *x* axis gives the magnitude relative to the real data; values smaller than 1 indicate a reduction in the expected counts. **(C)** Five sites with different shares. The larger the normalized entropy, the more equal the site shares are.

In order to rigorously analyze these factors while effectively controlling for other variables, we employ a semi-synthetic data analysis (see *"Semi-synthetic data analysis"* in the Materials and methods section) utilizing real daily COVID-related claims data. We first define the Poisson rate parameter for the simulation as the 7-day moving average of the real data, with the growth rate of the Poisson rate defined accordingly. Simulated observations are then generated by drawing Poisson-distributed samples from the rate parameter for each time step. The growth rate alert is then defined as whether the growth rate of the Poisson rate is larger than a threshold. Finally, we simulate the split of the sampled time series into a set of individual sites by drawing counts from a multinomial distribution. By varying the parameters of this distribution, we can control the degree of dispersion of data across sites. In the real world, the Poisson rates are not available even when the counts for all parties are known; thus the true growth rates and the correct growth rate alerts are unknown. For semi-synthetic data, the ground truth for each of these quantities is known, allowing us to compare how well different algorithms match it. In particular, we can distinguish between loss of accuracy due to the Poisson-distributed noise (reflected in the gap between the centralized method and ground truth) and loss due to decentralization of the data.

In Fig 6, each plot represents a comparison of the statistical powers of methods when the FDR is set at 0.10, in an analysis where one dimension is altered while all other dimensions are fixed. Fig 6A varies the number of sites while maintaining equal shares among them. Fig 6B scales the underlying Poisson rate parameter of the simulation by a multiplier while keeping the number of sites as $N = 8$, allowing us to examine performance with smaller counts where observations become sparser. The power of simply selecting the $p$-values from the largest site, represented by the green line, is not fully shown in the plot due to its significantly lower performance compared to the other methods. Fig 6C explores the case where the number of sites is fixed at $N = 5$, and the degree of imbalance in the shares is varied. The level of imbalance is quantified using the normalized entropy metric $S = (-\sum_{i=1}^{N} s_i \log s_i)/(\log N)$, where a value of 1 indicates perfectly equal shares. Of note, It's the imbalance of the share that substantially causes the difference of powers, rather than the entropy. We used the entropy only to evaluate the imbalance of the share after fixing the number of sites. A detailed elaboration of how we selected the shares to achieve these specified entropy values is provided in the *"Normalized entropy in semi-synthetic analysis"* section of S1 Text. We also include a three-panel heatmap displaying the relationship between entropy and the magnitude of counts, with one panel each for Stouffer's method, Fisher's method, and their difference, all under the fixed number of sites $N = 5$, in that section of S1 Text.

Our findings indicate that federated analysis performs well compared to the centralized setting. Relying solely on a single facility, even with a relatively large share of the counts, yields poor results. Fisher's method demonstrates the highest stability when the number of sites varies but each has an equal share of the total. Therefore, when the data is distributed among numerous sites, Fisher's method is the preferred choice for meta-analysis. Additionally, we observe that Fisher's method performs slightly better when the magnitudes of the counts are relatively small. In other cases, Stouffer's method performs better, perhaps reflecting the fact that the Gaussian approximation of the Binomial parameter is more accurate when the counts are larger. Furthermore, our analysis suggests that using only the largest site can outperform naive (unweighted) combination methods only when there is one dominant site in the entire region, as indicated by a normalized entropy of less than 0.7, which corresponds to the largest site having a share of at least 65%. For example, when the shares of five sites are (0.65, 0.1, 0.1, 0.1, 0.05), the normalized entropy of site shares is 0.70.

## Enhancing federated surveillance with auxiliary information

In the previous analysis of federated surveillance, we applied a meta-analysis framework which uses only $p$-values from the reporting sites. We might be able to use auxiliary information to further optimize the framework. Intuitively, hypothesis tests of larger sites are expected to be less noisy and should be weighed more than smaller sites. If the relative shares of the different sites are known even approximately, we can incorporate this information and assign weights to the different tests, as weighting is a common approach for integrating evidence [19,22–24].

In the Materials and methods section, we calculate the appropriate weighting scheme for different meta-analysis methods as a function of the relative shares of the reporting sites. Our framework differs from most previous meta-analysis studies [10,13,18–20] because the target can be reconstructed from the summation of decentralized counts when complete information is available. In contrast, in other meta-analysis studies, like Genome-wide association studies (GWAS), researchers combine the tests on whether a variant has an effect on a phenotype based on specific samples as the sample level information across different studies is unavailable. In such cases, the inverse of the standard error and the square root of the

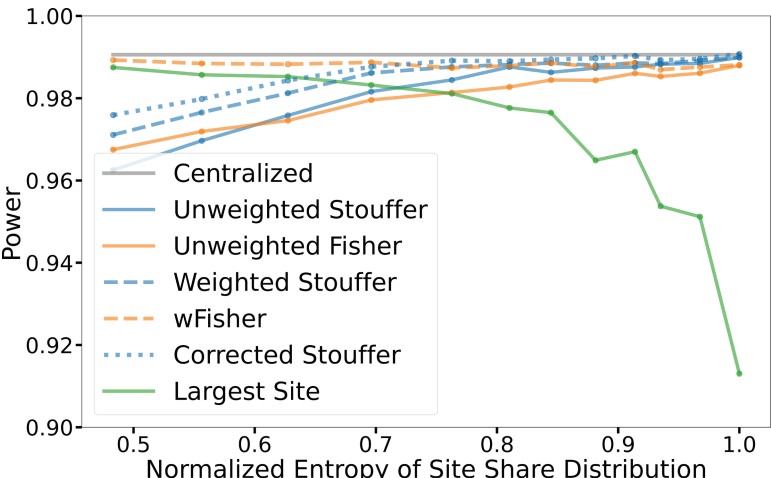

**Fig 7. Semi-synthetic analysis of the weighted methods.** Improvement with adaptive weighting and correcting. The larger the normalized entropy, the more equal the sites are.

sample size are suggested to be used as weights [18]. In our framework, we can explore different approximations of $p$-values and decide how to optimally weight them. For Stouffer's method, we show that weighting by the square roots of the shares recovers a Gaussian approximation to the centralized-data $p$-value. For Fisher's method, we compare several previously proposed weighting schemes [10,13,20] in simulation and observe that the wFisher method [10] performs best. In addition to adding weights, we can make a further modification to Stouffer's method when an estimate of the magnitudes of total reports of all sites is available. The modification involves incorporating a continuity correction term to make the results less conservative, which can be particularly useful when the total counts are small.

By comparing the performance of selecting the largest site with Stouffer's and Fisher's methods before and after incorporating weights, using our semi-synthetic data analysis framework, we observe in Fig 7 that after incorporating weights, both methods show improvements. Furthermore, the combined methods outperform selecting the largest site, even in the extreme setting where the largest site among all five accounts for 80% of the share. Specifically, the weighted Stouffer's method closely approximates the performance of the centralized setting when the shares between sites are similar, while wFisher demonstrates consistently superior performance when the shares of sites are unbalanced.

The performance of both the naive and weighted versions of Stouffer's and Fisher's methods is depicted in Fig 8A and 8B, where the weighted versions of both methods show improvement as anticipated. For HHS hospitalizations (Fig 8A), after weighting, Fisher's method's AUC increases from 0.93 to 0.94, with recall at precision of 0.90 increases from 0.71 to 0.77; Stouffer's method's AUC increases from 0.98 to 0.99, with recall at precision of 0.90 increases from 0.95 to 0.99. For outpatient insurance claims Fig 8B, after weighting, Fisher's method's AUC increases from 0.95 to 0.98, with recall at precision of 0.90 increases from 0.76 to 0.94; Stouffer's method's AUC increases from 0.87 to 0.93, with recall at precision of 0.90 increases from 0.65 to 0.84. Additionally, the inclusion of a continuity correction improves the AUC from 0.93 to 0.94 and recall at precision of 0.90 from 0.84 to 0.90 for daily-reported claim data with smaller counts. For weekly-reported hospitalization data, the counts are larger and the continuity correction has little impact.

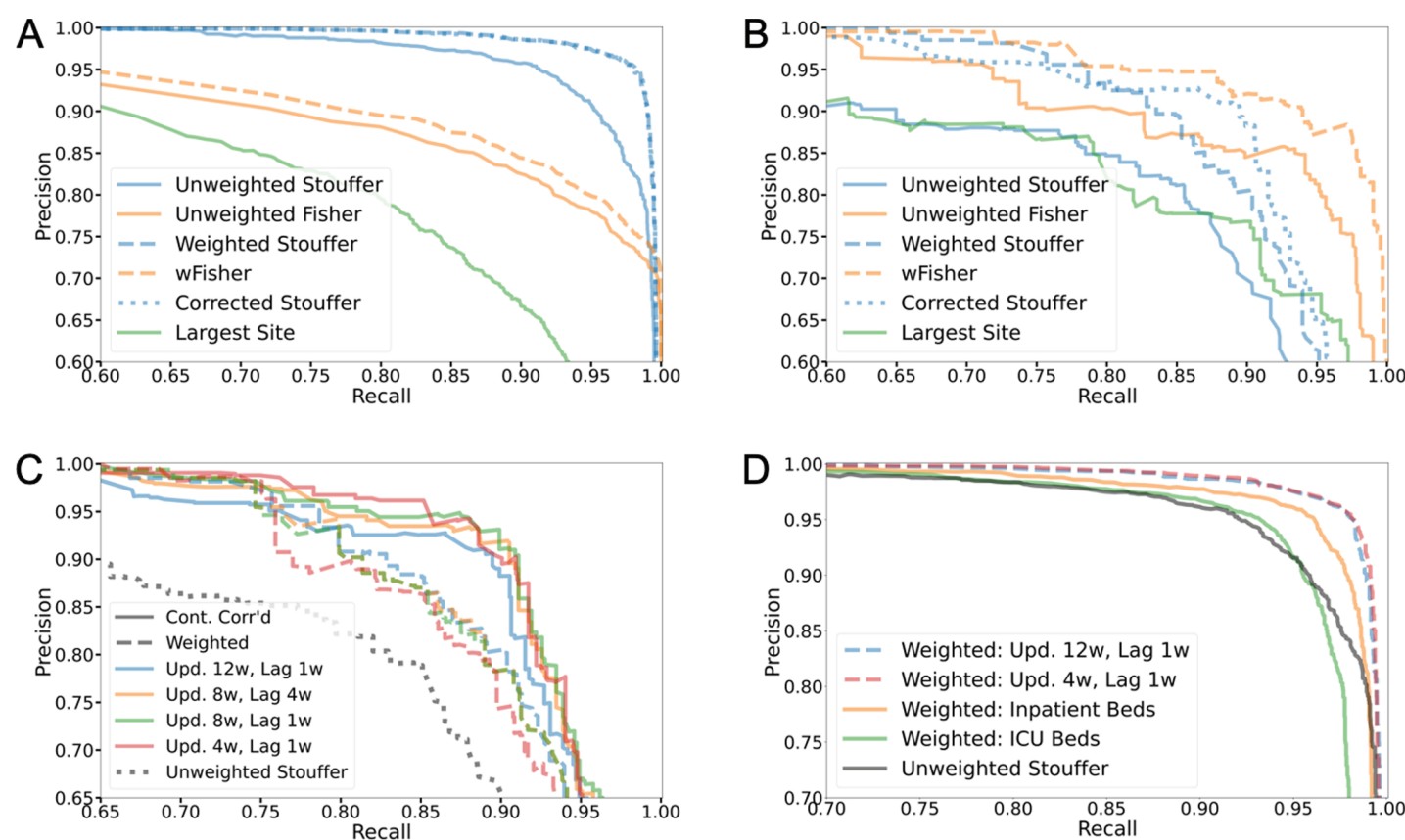

**Fig 8. Real data analysis of unweighted, weighted and continuity corrected methods. (A)** Facility-level, weekly, larger counts. **(B)** County-level, daily, smaller counts. **(C)** Weighted and continuity corrected Stouffer's method under different updating cycles (Upd.) and delays (Lag) of auxiliary information of claim data. **(D)** Weighted Stouffer's method using different auxiliary information of hospitalization data.

Finally, we test whether these patterns from the semi-synthetic experiments also hold in a more realistic setting. In the real world, the real-time shares of different sites may not be readily available, requiring the estimation of weights using auxiliary data. Various approaches can be employed depending on the available data. One potential setting is where sites report their total counts infrequently (e.g., monthly or quarterly), where they cannot be combined in real-time for surveillance, but may be used to estimate weights provided that the relative shares of different sites changes more slowly than the counts themselves. We test a range of possible reporting lags of auxiliary information used in weights and find that even relatively infrequent updates (e.g., weights estimated every 12 weeks) lead to improved performance with AUC around 0.93 compared to uniform weights with AUC = 0.87. The impact of the reporting cycle and lag of the auxiliary information on real datasets is illustrated in Fig 8C. Generally speaking, these factors have a small influence on the improvement achieved through weighted combinations. A second potential scenario is to use relatively static proxies for facility size to estimates weights, such as bed or ICU capacity. Empirically, we find that estimating the shares of providers using the bed or ICU usage generally improves performance, albeit with a marginal improvement shown here, compared to not having weights at all. However, this approach generally underperforms relative to methods that utilize specific information about facility-level COVID counts, despite those being delayed and sparsely updated, as detailed in Fig 8D.

## Discussion

Timely and accurate detection of outbreaks is critical to enable decision-making and responsive policy in public health emergencies. However, this task is difficult when critical data is distributed across multiple data custodians who may be unable or reluctant to share it. This study presents a framework for federated surveillance, which addresses these challenges by performing statistical analysis behind each data custodian's firewall followed by a meta-analysis to aggregate the evidence. Our results validate the potential for timely detection of emerging trends in population health without direct disclosure of any health data, even aggregated counts.

Our results also show the relative strengths and weaknesses of different meta-analysis methods under Poisson assumption for performing this aggregation on epidemiological data. We find that the relative performance of different p-value combination methods depends on the number of reporting sites, their relative sizes, and the expected magnitude of the counts. Stouffer's method performs best where data is concentrated in a smaller number of sites and the magnitude of reports is relatively large. On the other hand, Fisher's method exhibits robustness in more challenging settings characterized by a larger number of data holders and greater imbalances of shares among them. The inclusion of additional information, such as the sites' shares and estimated total counts within a given region, enables additional improvements in performance. Across all settings and datasets that we consider, we find that at least one meta-analysis method results in statistical power that closely approximates the best attainable if all data were available for a single analysis. Our experiments primarily focus on the Poisson rate ratio test, yet the conclusions drawn are not confined to this context alone. With minor modifications to the framework, similar results can be achieved for other hypothesis tests (such as the Poisson test itself or under different data distribution assumptions, including the (Log) Normal distribution which uses Gaussian approximation of sufficiently large counts, or overdispersed Poisson and Negative Binomial distribution, which accounts for additional over-dispersion [25]. Addressing the temporal patterns of reporting delays such as backfill may further increase power.

Federated surveillance provides a simple, readily implementable framework for addressing the practical barriers to including already-existing health system data in public health surveillance systems. While more complex methodologies such as homomorphic encryption could provide stronger theoretical guarantees, the methods presented here are more readily understood by the lay public and hence more likely to be acceptable to health data custodians. Implementation will likely rely on mandatory reporting guidelines, alongside easily-usable software for agreed-upon hypothesis tests, not limited only to Poisson counts. Our work demonstrates that relatively simple meta-analysis methods can enable significantly more accurate and timely warnings of changes in population health without the creation of centralized datasets.

## Supporting information

**S1 Appendix:** Contains detailed sections on: Parameters and Notations; Power Computation; Power Curves and Calibrations; Normalized Entropy in Semi-synthetic Analysis; Proof for Equation 10; Summary of Ground Truth Windows;Detection Delay.
(PDF)

## Acknowledgments

This material is based upon work supported by the United States of America Department of Health and Human Services, Centers for Disease Control and Prevention, under award number NU38FT000005; and contract number 75D30123C1590. Any opinions, findings, and conclusions or recommendations expressed in this material are those of the author(s) and do not necessarily reflect the views of the United States of America Department of Health and Human Services, Centers for Disease Control and Prevention.

## Author contributions

**Conceptualization:** Ruiqi Lyu, Roni Rosenfeld, Bryan Wilder.

**Formal analysis:** Ruiqi Lyu.

**Investigation:** Ruiqi Lyu.

**Methodology:** Ruiqi Lyu.

**Software:** Ruiqi Lyu.

**Supervision:** Roni Rosenfeld, Bryan Wilder.

**Writing – original draft:** Ruiqi Lyu.

**Writing – review & editing:** Ruiqi Lyu, Roni Rosenfeld, Bryan Wilder.

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
