## [Decision Letter · Decision Letter 0]

4 Nov 2024

PCOMPBIOL-D-24-01600Federated Epidemic SurveillancePLOS Computational Biology Dear Dr. Lyu, Thank you for submitting your manuscript to PLOS Computational Biology. After careful consideration, we feel that it has merit but does not fully meet PLOS Computational Biology's publication criteria as it currently stands. Therefore, we invite you to submit a revised version of the manuscript that addresses the points raised during the review process. Please submit your revised manuscript within 60 days Jan 04 2025 11:59PM. If you will need more time than this to complete your revisions, please reply to this message or contact the journal office at ploscompbiol@plos.org. Please include the following items when submitting your revised manuscript: * A rebuttal letter that responds to each point raised by the editor and reviewer(s). You should upload this letter as a separate file labeled 'Response to Reviewers'. This file does not need to include responses to formatting updates and technical items listed in the 'Journal Requirements' section below.* A marked-up copy of your manuscript that highlights changes made to the original version. You should upload this as a separate file labeled 'Revised Manuscript with Track Changes'.* An unmarked version of your revised paper without tracked changes. You should upload this as a separate file labeled 'Manuscript'. If you would like to make changes to your financial disclosure, competing interests statement, or data availability statement, please make these updates within the submission form at the time of resubmission. Guidelines for resubmitting your figure files are available below the reviewer comments at the end of this letter. We look forward to receiving your revised manuscript. Kind regards, Benjamin AlthouseSection EditorPLOS Computational Biology Virginia PitzerSection EditorPLOS Computational Biology Feilim Mac GabhannEditor-in-ChiefPLOS Computational Biology Jason PapinEditor-in-ChiefPLOS Computational Biology  **Journal Requirements:** **Additional Editor Comments (if provided):****Reviewers' comments:** Reviewer's Responses to Questions

**Comments to the Authors:**

Reviewer #1: See comments in the attached PDF.

I left a few comments throughout. Most are minor edits for clarity. I have two main points that can be addressed with minor revisions.

First, this work is only relevant if implemented. It can only be implemented if the research community gets buy-in from data custodians or if mandatory reporting guidelines are shifted to include p-values for certain targets. Both of these are big tasks. Could you write to that in the discussion? Maybe you've thought of additional ways to get this data, but as you well know, it's hard to get raw data from facilities for all of the reasons you mentioned in the paper. I can't imagine that many administrators are going to be amenable to running some quick stats to send to researchers. Do you have any ideas about how to get a system like this up and running?

Second, could you include a two-panel heatmap of entropy vs. magnitude of counts, with one panel for Stouffer and one for Fisher (even if it needs to be in a supplement)? I think the unweighted methods are probably sufficient. There are more details in the comments in the PDF.

I did read the supplement, but I have no comments or suggestions.

Reviewer #2: This is a well written and comprehensive look at an important and novel area of epidemic science. My overall impression is that this manuscript covers a lot of ground, and could benefit from moving some sections to supplemental materials (eg deep dives on methods: pages 11-14) . Additionally, an architectural diagram of the abstract process (or one that is more detailed describing the different study designs) would help a bit, as there are a lot concepts to entertain and only detailed plots of data to guide the reader. If not already planned a companion article describing the general proposed solution and its applications with less focus on methodology would be very useful to the community writ large.

As is, this serves as a good anchor piece to describe the concept, detail the methodologies, and define a framework for evaluation with results. I would urge the authors to consider reorganizing some of the text or provide more overall structure (to the degree possible within the journal’s formatting requirements).

Major comments

• I understand that this is intended to be abstract and applicable to a wide range of possible applications. However, a lot of the paper is spent on evaluation (as it should be). I am quite familiar with the datasets described, however, it is not exactly clear the overall size of the datasets (eg number of observations, total number of facilities etc) used in these analyses. Figure 2 provides one perspective of the data, but still lacks some of these simple qualitative descriptions. For instance how many aggregate level “true” surges are there?

• Embedded in the detailed description of the methodology is that the main task is to detect a surge at aggregate levels based on the p-values of the surge detection statistic at the individual sites. I assumed, incorrectly, based on the datasets, that this would be state level from facilities and state level from counties (HHS and Change healthcare claims respectively). It’s not till page 15 in the description of the data generation that the word “state” is used, and then on page 18 the scale and level of the task comes into focus when the exact comparison is spelled out: “we add up the counts of facility-level hospitalization to generate county-level p-value alerts, and county-level insurance claim counts to create state-level p-value alerts”

Perhaps an overview of the proposed evaluations earlier on in the materials and methods would benefit other readers (this would be more clear too with quick description of the data suggested above as well). I think this would also help orient readers to the various stages of the evaluation. As is, the authors provide a nice narrative and gently guide the reader through the successive levels of evaluation, however, given the depth and length a summary up front would be helpful.

• In the evaluation of the surge detection task section there is an important assumption about what is “true” vs. “false” positive in lines 324-328. A one month window seems generous, though in practice 2 weeks after is still useful especially if true ground truth data is not knowable, so I think this is “reasonable.” Interpretation of the findings would benefit from understanding the proportion of all time points that are covered by these windows around true surges vs. not covered. A distribution of how many of these “true positives” come 1 week early vs. 2 weeks later is interesting as well. There are a lot of sensitivities explored in this paper however the impact of this definition seems important and seems to be left unexplored.

• Review of the code helpfully provided on github with embedded data was useful, however with the limited documentation I was not able to get the scripts to run. I imagine there is a specific order that the analyses build off of and some dependencies that I may not have (though none I initially saw). Some of the errors suggested that there may be some directory structures hard-coded that would make this not possible to run outside of the author’s personal environment (eg plot_amp_rewrite.py) Some documentation about what each script does and/or protocol is needed to make these results reproducible.

Minor comments:

• Lines 191-3: Not sure how this last sentence fits in.

• Line 210: starts with an unnecessary (?) indentation

• Line 351: The “Change” dataset is referenced here for the first time since line 57 and given it’s name is a common word perhaps clarifying with “Change healthcare” would be useful. Took me several readings to remember this unusually named data source.

• Line 416: This would be a good place to remind the reader how many observations are contained in this specific date range.

• Lines 417-8: From review of the data in the github repository, I think you are adding up facilities for a metro-area level alert (not county as stated here). Stating this earlier in the methods as detailed in the major comments would be helpful.

• Lines 423-5: This is stated many times and is important. Details on how many sites and importantly “how they were chosen” should be included somewhere. I am looking at PA_Scranton.csv in the data repo and don’t see facility 390119, which is the example facility on the delphi page for this dataset. Why was this facility not included when its address is in Scranton?

• Lines 434-437: The case of a single facility is described before the motivation of its inclusion (lines 448-9) which is a little awkward.

• Line 503: … perhaps reflecting THE fact that….

• Figure 7d: When printed the overlap of the blue and red dashed lines looked purple which was initially confusing, perhaps only one is needed.

**Have the authors made all data and (if applicable) computational code underlying the findings in their manuscript fully available?**

Reviewer #1: Yes

Reviewer #2: Yes

PLOS authors have the option to publish the peer review history of their article (what does this mean?). If published, this will include your full peer review and any attached files.

Reviewer #1: No

Reviewer #2: **Yes: **Bryan Lewis

 **Figure resubmission:**While revising your submission, please upload your figure files to the Preflight Analysis and Conversion Engine (PACE) digital diagnostic tool, https://pacev2.apexcovantage.com/. PACE helps ensure that figures meet PLOS requirements. To use PACE, you must first register as a user. Registration is free. Then, login and navigate to the UPLOAD tab, where you will find detailed instructions on how to use the tool. If you encounter any issues or have any questions when using PACE, please email PLOS at figures@plos.org. Please note that Supporting Information files do not need this step. If there are other versions of figure files still present in your submission file inventory at resubmission, please replace them with the PACE-processed versions. 
---

## [Decision Letter · Decision Letter 1]

24 Feb 2025

Dear Ms. Lyu,

We are pleased to inform you that your manuscript 'Federated Epidemic Surveillance' has been provisionally accepted for publication in PLOS Computational Biology.

Best regards,

Benjamin Althouse

Section Editor

PLOS Computational Biology

Virginia Pitzer

Section Editor

PLOS Computational Biology

Reviewer's Responses to Questions

**Comments to the Authors:**

Reviewer #2: Thank you for your thorough look at comments and the changes you've rendered. I found the sensitivity on the width of the "true positive" window quite compelling. The performance seems to work well.

I endeavored to run the code again and encountered some minor issues documented in the attached document.

**Have the authors made all data and (if applicable) computational code underlying the findings in their manuscript fully available?**

Reviewer #2: Yes

PLOS authors have the option to publish the peer review history of their article (what does this mean?). If published, this will include your full peer review and any attached files.

Reviewer #2: **Yes: **Bryan Lewis

---

## [Editor Report · Acceptance letter]

PCOMPBIOL-D-24-01600R1

Federated Epidemic Surveillance

Dear Dr Lyu,

I am pleased to inform you that your manuscript has been formally accepted for publication in PLOS Computational Biology. Your manuscript is now with our production department and you will be notified of the publication date in due course.

With kind regards,

Anita Estes
